# An exploratory study of outpatient medication knowledge and satisfaction with medication counselling at selected hospital pharmacies in Northwestern Nigeria

Samirah N. Abdu-Aguye[1]*, Kamilu S. Labaran[1], Nuhu M. Danjuma[2], Shafiu Mohammed[1,3]

1 Department of Clinical Pharmacy & Pharmacy Practice, Ahmadu Bello University Zaria, Zaria, Kaduna State, Nigeria, 2 Department of Pharmacology & Therapeutics, Ahmadu Bello University Zaria, Zaria, Kaduna State, Nigeria, 3 Health Systems and Policy Research Unit, Ahmadu Bello University Zaria, Zaria, Kaduna State, Nigeria

☯ These authors contributed equally to this work.

* sn.abduaguye@gmail.com

**Data Availability Statement:** All relevant data files are available from the OPENICPSR database:

## Abstract

### Background

Medication counselling is an important activity that improves patient therapeutic outcomes. After this activity has been carried out, patients should be satisfied with counselling, and possess adequate knowledge about their medications.

### Objectives

To describe outpatient/caregiver medication knowledge and satisfaction with medication counselling at the main outpatient pharmacies of eight public secondary and tertiary hospitals located in two states in Northwestern Nigeria.

### Methods

Exit interviews were conducted from December 2019 to March 2020 with randomly sampled patients/caregivers who had just been dispensed one or more prescription medications from the main pharmacies of the hospitals. The questionnaire used contained 31 questions in three sections. The first section collected demographic information. The second section assessed respondents' experiences and overall satisfaction with the counselling they had received. The last section evaluated respondents' knowledge of one randomly selected prescription medication that had been dispensed to them. Data collected were coded and analyzed to generate descriptive statistics. To explore associations between respondent characteristics and overall satisfaction, non-parametric tests were used, and statistical significance set at $p < 0.05$.

https://www.openicpsr.org/openicpsr/project/159601.

**Funding:** The author(s) received no specific funding for this work.

**Competing interests:** The authors have declared that no competing interests exist.

## Results

A total of 684 patients/caregivers were interviewed. Majority of respondents agreed that the time spent (97.1%) and quantity of information (99.1%) provided during counselling was adequate. However, over 60% of them also agreed that dispensers did not assess their understanding of information provided or invite them to ask questions. Despite this, their average overall satisfaction with counselling on a 10-point scale was 8.6 ± 1.6. Over 90% of them also correctly identified the routes and frequency of administration of the prescribed medication selected for the knowledge assessment. Although, more than 60% of respondents did not know the duration of therapy or names of these medications.

## Conclusion

Respondents' satisfaction with medication counselling was fairly high even though they did not seem to know much about their medication.

## Introduction

Medication counselling is the provision of verbal or written information about medicines to individual patients or their caregivers. It is a routine, but very important activity usually carried out by pharmacists or related personnel in healthcare settings including hospital pharmacies.

Medication counselling has several benefits for patients. Since proper counselling enhances patient medication knowledge [1, 2], it can also prevent the occurrence of adverse drug events [3], improve patient medication adherence [4, 5], increase patient satisfaction with care [6, 7], and improve patient beliefs about their medicines [8]- all of which culminate in better therapeutic outcomes for patients. Studies have also reported that when patients are satisfied with the medication counselling they have received, they are more likely to adhere and use their medications correctly [9, 10].

After medication counselling has been completed, patients ideally should be satisfied with the counselling they have received, and also possess adequate knowledge about their medications. However, several studies from all around the world that have assessed outpatient medication knowledge following medication counselling, have generally reported that their respondents had poor knowledge about at least one aspect of their medication [7, 11–17]. Similarly, the few studies [7, 18] that have evaluated satisfaction with medication counselling have also reported varying levels of patient satisfaction.

Nigeria like several other low-middle income countries, has a sub-optimal healthcare system [19]. Primary healthcare centers within the country are mostly non-functional, in addition to suffering from other issues including poor availability of qualified staff, essential medical supplies and medicines [20]. As a result of this, many individuals living in the country bypass primary health facilities and move to secondary or tertiary hospitals for all their healthcare needs [21]. Thus, most public secondary and tertiary hospitals within the country have specially designated units called "outpatient or family medicine" departments that handle routine medical complaints from members of the public [21]. These hospitals also have one or more pharmacies located inside them that dispense prescriptions generated from these outpatient departments/clinics. These hospital pharmacies are usually manned by trained pharmacists and pharmacy technicians, all of whom are expected to counsel patients after filling their

prescriptions. Although, it should however be noted that there are no explicit guidelines, policies, incentives or standards for medication counselling within the country.

Little is known about outpatient medication knowledge and satisfaction with medication counselling within the Nigerian setting [22, 23], thus the need for further research on these topics. Furthermore, because medication knowledge and satisfaction with counselling are linked to the optimal use of medication by patients, assessing these variables are important steps in evaluating the quality of services provided and identifying areas for improvement. Thus, the aim of this study was to describe outpatient/caregiver medication knowledge and satisfaction with medication counselling provided at outpatient pharmacies of selected hospitals located in North-Western Nigeria.

## Methods

### Study sites, design and population

Exit interviews were conducted from December 2019 to March 2020 on patients/caregivers who had just been dispensed one or more prescribed medication from the major outpatient pharmacies of eight public secondary and tertiary hospitals located in two states (Kaduna and Kano) in Northwestern Nigeria. The tertiary hospitals included Ahmadu Bello University Teaching Hospital Zaria, Aminu Kano Teaching Hospital Kano, Barau Dikko Teaching Hospital Kaduna, and Mohammed Abdullahi Wase Teaching Hospital Kano. The secondary hospitals included Yusuf Dantsoho Memorial Hospital Kaduna, Gwamna Awan General Hospital Kaduna, Murtala Mohammed Specialist Hospital, Kano and Hasiya Bayero Pediatric Hospital Kano. To be included in the study, respondents had to consent to participate, be outpatients, be aged at least 16 years, and understand either English or Hausa (English is the official language of Nigeria, while Hausa is the predominant ethnic language spoken in Northern Nigeria).

### Sample size determination and sampling technique

Despite being called 'outpatient' pharmacies, all the selected pharmacies also attended to hospital inpatient prescriptions, and all prescriptions were recorded in the same record books. This ensured that it was impossible to differentiate between previous inpatient and outpatient prescription records for the purposes of sample size calculation. Thus, it was not possible to accurately calculate sample sizes for only outpatient visits-which were the focus of this study.

Consequently, non-proportional quota sampling was used to ensure the same level of representation and a quota of 100 exit interviews was allocated to each pharmacy (yielding an estimated total sample size of 800). Respondents were randomly sampled for these interviews.

### Data collection instrument

A questionnaire was designed for the study. It contained 31 questions distributed into three sections. The first section contained 12 questions that collected information about demographic and other characteristics of respondents including gender, age, number of drugs prescribed etc.

The second section contained 11 questions, some of which were adapted from other studies [24–26]. Ten of these questions assessed respondents' experiences during the medication counselling process and were answered using the five point Likert scale. The last question in this section assessed respondents' overall satisfaction with the counselling they had received, and this was answered using a global 10-point rating scale. The items in this section were sent to 11 purposively sampled Clinical Pharmacy experts to ascertain their content validity. The

calculated average content validity index (Av-CVI) score of the items was 0.96, and no item scored below 0.82, which is considered appropriate [27]. These questions were then pretested on a sample of 103 outpatients at a separate secondary hospital (not one of the study sites). After the pilot test, the internal reliability coefficient (Cronbach's alpha) was calculated and found to be 0.7, which is also considered acceptable [28]. Thus, no adjustments were carried out.

The last section of the questionnaire contained eight questions that aimed to assess respondents' knowledge of prescribed medication that had been dispensed to them. These open-ended questions were adapted from those used in earlier studies by Okuyan *et al.*, [4] and Hirko *et al.* [16].

## Data collection

Data was collected in each hospital over the course of one week (Monday-Friday) from 9 am to 2pm daily. Three young pharmacists with 1–2 years' work experience served as the research assistants who collected the data. They were trained prior to data collection over a three-day period by the principal investigator, who also sat through the first 20 interviews conducted by each of these assistants. Patients/ caregivers were randomly approached by these assistants and asked whether they were outpatients who had just collected prescribed medication from the pharmacy. If they answered "yes", they were then invited to participate-after the study objective had been briefly explained to them. If they were willing to participate, they were invited to sit down, and the questionnaire (in their preferred language) used to interview them. For the knowledge assessment questions, the patient was asked to randomly select one medicine from the bag containing their prescribed medication and hold the selected drug all through the duration of questioning. The generic name of the medication, its duration of use and frequency of administration were then noted down on the questionnaire by the data collector, after which respondents' medication knowledge was assessed. Their answers were written down verbatim. The exit interview sessions mostly lasted between 6–15 minutes.

## Data analysis

Data collected were coded and entered into the IBM Statistical Package for the Social Sciences (SPSS) version 22 software and analyzed to generate descriptive statistics (frequencies and percentages). For the responses to the questions in section two that were answered using a five-point Likert scale, respondents who "strongly agreed" and "agreed" were grouped and reported together and the same was done for respondents who "strongly disagreed" and "disagreed".

Medications used to assess respondents' medication knowledge were categorized into groups based on the Anatomical Therapeutic Chemical (ATC) classification system developed by the World Health Organization (WHO) Collaborating Center for Drug Statistics Methodology in Norway [29].

Respondents' answers to the open-ended medication knowledge questions were assessed using a conference format after data collection had been completed by four pharmacists (2 academic and 2 hospital pharmacists) with previous experience in similar studies. Before a respondents' answer could be classified as correct or wrong, 3 out of 4 of the assessors had to agree or disagree. Answers to the questions on duration and frequency of administration were assessed by cross-checking with the information contained on their prescriptions that had been earlier copied out by the data collectors. Where this was not available and for all other questions including medication indication and additional information relevant to the drug, the researchers used their prior knowledge of pharmacology and pharmacotherapeutics (and cross-checked with various reference sources where necessary) to assess respondent

knowledge. Finally, for the question on what to do if a dose was missed, respondent answers were grouped into two themes to simplify reporting.

To calculate overall medication knowledge, correct responses to the medication knowledge assessment questions were scored one mark each, while wrong and "I don't know" responses were scored zero. For selected medicines (e.g., stat dose medications) where respondents did not require certain information, those components were also scored 1 mark. All these responses were then totaled to produce a score /8 for each participant. Respondents who scored 5 or more were classified as having good knowledge, while those who scored 4 or less were classed as having poor knowledge.

To explore associations between respondent characteristics and their overall satisfaction, non-parametric tests (Mann Whitney U and Kruskal-Wallis) were used, and statistical significance set at $p < 0.05$. In the cases where statistical significance differences in overall satisfaction scores between groups was observed, Dunn-Bonferroni post hoc tests were also carried out to identify the specific subgroups involved.

## Ethics approval

Ethical clearance for the study was obtained from the ethical review committees of Kaduna State Ministry of Health and Human Services (MOH/ADM/744/VOL.1/723), Kano State Ministry of Health (MOH/Off/797/T.I./1807), Ahmadu Bello University Teaching Hospital (ABUTHZ/HREC/G30/2019), Aminu Kano Teaching Hospital (NHREC/28/01/2020/AKTH/EC/2808), Barau Dikko Teaching Hospital (19-0004-11) and from Ahmadu Bello University, Zaria (ABUCUHSR/2020/017). Verbal consent from respondents was considered to be informed consent, and each participant was asked to provide consent before the interviews were conducted

## Results

A total of 684 patients/caregivers were interviewed, producing an 85.5% total response rate. The sample size quota (100 patients) for all four hospitals located in Kano state was achieved, while 57, 69, 76 and 82 patients were interviewed from the four hospitals located in Kaduna state.

### Demographic and other characteristics of respondents

Demographic and other characteristics of these individuals are reported below in Table 1. The ages of study participants ranged from 16–78 years (average = 35.3 years), and over half of them were females (Table 1).

### Respondents' experiences during medication dispensing & counselling

Majority of study respondents agreed that the time spent and quantity of information provided during counselling was adequate (Table 2). Most of them also agreed that dispensers were friendly and used language(s) or terms that they could understand during counselling. On the other hand, over 60% of respondents agreed with the statements that dispensers did not assess their understanding of the information they provided or invite them to ask questions.

### Respondents' overall satisfaction with medication counselling

When respondents were asked to rate their overall satisfaction with medication counselling on a scale of one to ten, only 27 respondents (4%) provided ratings of 5 or lower. The average

**Table 1. Demographic and other characteristics of exit interview respondents (n = 684).**

| Characteristic | Variables | n (%) |
|---|---|---|
| *Gender* | Female | 445 (65.1) |
| | Male | 239 (34.9) |
| *Highest educational level completed | No formal education | 127 (18.7) |
| | Primary school | 47 (6.9) |
| | Secondary school | 251 (36.9) |
| | Two years of tertiary education | 119 (17.5) |
| | Four or more years of tertiary education | 137 (20.2) |
| *Monthly income | NGN 18,000 or less | 43 (14.5) |
| | NGN 18,001–50,000 | 170 (57.2) |
| | NGN 50,001–100,000 | 58 (19.5) |
| | Above NGN 100,000 | 26 (8.8) |
| *Owner of prescription* | Self | 397 (58) |
| | Other | 287 (42) |
| *Nature of area where counselling was provided* | Private[a] | 74 (10.8) |
| | Semi-private[b] | 410 (60) |
| | Window[c] | 200 (29.2) |

NGN-Nigerian Naira. 1 US Dollar = 360 NGN at the time of data collection *Values in these rows sum up to less than the total because of missing values.

[a]Private = Counselling area is secluded and has a door that can be shut, both dispensers and patients can sit comfortably.

[b]Semi-private = Patient can come into the pharmacy and may sit down, however the area is not secluded and conversations can be overheard.

[c]Window = Patients cannot enter the pharmacy, and dispensers communicate with patients through a window.

overall satisfaction score was 8.6 ± 1.6, and a marked ceiling effect was observed as 282 respondents (41.5%) rated their satisfaction as 10/10.

## Associations between respondent characteristics and overall satisfaction with medication counselling

Associations between respondent characteristics and overall satisfaction with medication counselling scores are shown below in Table 3. Overall satisfaction scores were significantly higher in

**Table 2. Respondents experiences during medication counselling (n = 683).**

| Item | Agreed n (%) | Neutral n (%) | Disagreed n (%) |
|---|---|---|---|
| *Waiting time in the pharmacy was too long* | 88 (12.9) | 58 (8.5) | 537 (78.6) |
| *Time spent by the dispenser during counselling was adequate | 662 (97.1) | 11 (1.6) | 9 (1.3) |
| *Place where counselling took place was comfortable | 650 (95.7) | 13 (1.9) | 16 (2.4) |
| *Friendliness of the dispenser was poor* | 17 (2.5) | 14 (2.1) | 648 (95.4) |
| *Privacy of counselling area was adequate | 524 (77) | 69 (10.1) | 88 (12.9) |
| *Unfamiliar medical terms were used during counselling | 12 (1.8) | 8 (1.2) | 662 (97) |
| *Language used by the dispenser was understandable* | 662 (96.9) | 1 (0.2) | 20 (2.9) |
| *Dispenser did not assess respondents' understanding of the information provided* | 459 (67.2) | 26 (3.8) | 198 (29) |
| *Dispenser invited respondent to ask questions during counselling | 221 (32.4) | 21 (3.1) | 439 (64.5) |
| *Quantity of information about respondents' medicines provided by the dispenser was adequate* | 677 (99.1) | 4 (0.6) | 2 (0.3) |

*Values in these rows sum up to less than the total because of missing values.

**Table 3. Associations between respondent characteristics and overall satisfaction with medication counselling scores.**

| Characteristic | Variables | Mean rank | p value |
|---|---|---|---|
| *Gender* | Female | 356.8 | 0.001*a |
| | Male | 308.8 | |
| *Highest educational level completed* | No formal education | 408.4 | < 0.001*b |
| | Primary school | 337.1 | |
| | Secondary school | 353.4 | |
| | 2 years of tertiary education | 298.7 | |
| | 4 or more years of tertiary education | 280.8 | |
| *Monthly income* | NGN 18,000 or less | 193.5 | <0.001*b |
| | NGN 18,001–50,000 | 147.2 | |
| | NGN 50,001–100,000 | 132.1 | |
| | Above NGN 100,000 | 101.0 | |
| *Nature of area where counselling was provided* | Private | 332.1 | < 0.001*b |
| | Semi-private | 213.9 | |
| | Window | 402.0 | |

NGN-Nigerian Naira, 1 US Dollar = 360 NGN at the time of data collection

*Significant at p< 0.05.

a-Mann-Whitney U test

b-Kruskall Wallis H test.

female respondents when compared to those of male respondents (p = 0.001). Satisfaction scores also decreased as the educational level and income of respondents increased. The Dunn-Bonferoni post hoc test revealed that there were statistically significant differences in the overall satisfaction scores between respondents with no formal education and those with either 2 or 4 years of post-secondary education (p < 0.001 in both cases). Similarly, there was also a statistically significant difference between the scores of respondents that had completed secondary education and those who had completed four or more years of post-secondary education (p = 0.003).

In the same vein, as respondents' income increased, their overall satisfaction with medication counselling also decreased. There were statistically significant differences in the satisfaction with medication counselling ratings of respondents earning 18,000 NGN or less monthly and those earning more. Furthermore, there was also a statistically significant difference in the satisfaction scores of those earning between 18,000–50,000 NGN and those earning over 100,000 NGN monthly (p = 0.042).

## Respondents' medication knowledge

Respondents' responses to the questions asked to assess their knowledge of the selected medications is reported below in Table 4. Over 90% of them correctly identified the routes, dose and dosing frequency of the selected medication (Table 4). Less than half of them knew the correct indication (48.3%) and daily timing (47.1%) for these medications. Over 60% of them also did not know the duration of therapy, name or any other additional information about these medications (Table 4). Furthermore, only 261 respondents (38.2%) had good medication knowledge, as defined by an overall total medication knowledge score of 5 or higher.

## Respondents' medication knowledge by ATC category

Respondents' knowledge of the four most common ATC classes of medication assessed are shown below in Table 5.

**Table 4. Responses to the questions assessing respondents medication knowledge (n = 684).**

| | Participants' Response | n (%) |
|---|---|---|
| *Name of medication* | Correctly provided the medications' generic name | 74 (10.8) |
| | Correctly provided a brand name for the medication | 103 (15.1) |
| | Provided a wrong generic/brand name | 3 (0.4) |
| | Did not know | 504 (73.7) |
| *Indication for the medication* | Provided a correct answer | 330 (48.3) |
| | Provided a wrong answer | 58 (8.5) |
| | Did not know | 296 (43.3) |
| *Route of administering the medication* | Provided the correct answer | 683 (99.9) |
| | Provided the wrong answer | 1 (0.1) |
| *Dose and frequency of administration of the medication* | Provided the correct answer | 641 (93.7) |
| | Provided a wrong answer | 38 (5.6) |
| | Did not know | 5 (0.7) |
| *Duration of use for the medication* | Provided the correct answer | 214 (31.3) |
| | Provided a wrong answer | 18 (2.6) |
| | Did not know | 433 (63.3) |
| | Not required [a] | 19 (2.8) |
| *Daily timing of use for the medication* | Provided the correct answer | 322 (47.1) |
| | Provided a wrong answer | 42 (6.1) |
| | Did not know | 301 (44) |
| | Not required [a] | 19 (2.8) |
| *[*]What to do if a dose of the medication was missed* | Respondents said they would not combine two doses at once | 264 (38.7) |
| | Respondent said they would take as soon as they remembered | 23 (3.3) |
| | Did not know | 375 (54.9) |
| | Not required [b] | 21 (3.1) |
| *Any other additional information provided about the medication* | Provided a correct answer | 54 (7.9) |
| | Provided a wrong answer | 2 (0.3) |
| | Did not know | 591 (86.4) |
| | Not required [c] | 37 (5.4) |

[*]Values in this row sum up to less than the total because of missing values.

[a] This answer was not required for single use or stat doses of some drugs e.g. Albendazole, Fluconazole and Azithromycin.

[b] This answer was not required for single use or stat doses of some drugs and dermatological preparations.

[c] This answer was not required for single use or stat doses of some drugs and selected vitamin preparations including Vitamins B complex and C.

**Table 5. Percentage of respondents with correct responses to questions assessing their medication knowledge of drugs in selected ATC categories.**

| Subgroup | Name[a] | Indication[b] | Duration[c] | Daily timing[c] | What to do if a dose was missed[d] | Additional information[c] |
|---|---|---|---|---|---|---|
| *Drugs acting on the cardiovascular system ATC Category C (n = 77)* | 15.6% | 77.6% | 16.9% | 57.1% | 35.1% | 9.1% |
| *Systemic anti-infective drugs ATC Category J (n = 222)* | 23% | 20.3% | 34.7% | 26.6% | 46.1% | 14.9% |
| *Drugs acting on the musculoskeletal system ATC Category M (n = 42)* | 14.3% | 50% | 35.7% | 64.3% | 42.9% | 4.8% |
| *Anti-parasitic Products ATC Category P (n = 74)* | 20.3% | 73% | 70.3% | 40.5% | 40.5% | 6.8% |

[a]Correct for name = Respondent provided a correct brand or generic name.

[b]Correct for indication = Respondent provided a correct possible indication for use of the drug.

[c]Correct for duration, daily timing and additional information = Respondent provided the correct answer or that information was not required for the medication.

[d]Correct for what to do if dose was missed = Respondent knew they were not supposed to double medication dose or they said they would take as soon as they remembered.

The 77 medications examined from the ATC category C class included diuretics, beta-blockers, calcium channel blockers, angiotensin converting enzyme inhibitors/angiotensin receptor blockers or fixed dose combinations of these agents. Over half of these respondents knew the correct daily timing (57.1%) and indications (77.6%) for these agents (Table 5).

The 222 medications examined from the ATC category J class included various types of antibiotics from different classes, fixed dose combinations of antibiotics and systemic antifungal drugs. Less than half of these respondents knew the correct duration of drug therapy (34.7%), daily timing (26.6%) and indication for use (20.3%) for these agents (Table 5).

Most of the 42 medications examined from the ATC category M class contained non-steroidal anti-inflammatory drugs (NSAIDs) either singly or in combination with other agents. Half or more these respondents knew the correct daily dose timings (64.3%) and indication (50%) for these drugs (Table 5).

The 74 medications examined from the ATC category P class included various types of anti-malarials including Artemisinin Combination Therapy (ACT) drugs, Sulphadoxine/Pyrimethamine, and other antiparasitic drugs like albendazole (Table 5). Over half of these respondents knew the correct duration of drug therapy (70.3%), and drug indication (73%).

## Discussion

This study described patient/caregiver medication knowledge and satisfaction with medication counselling. Study findings showed that the average overall satisfaction with medication counselling score was 8.6 on a 10-point scale. Majority of the patients' or caregivers interviewed agreed that the time spent and quantity of information provided during counselling were adequate, and that pharmacy waiting times were not too long. Over half of them also agreed that dispensers did not assess their understanding of the information provided or invite them to ask questions during counselling. Most of them correctly identified the routes and frequency of administration of the medicines they had selected for the knowledge assessment, although more than 60% of them had no knowledge of the duration of therapy, name or any other additional information about these medications.

With respect to the experiences of this study's respondents' during medication counselling, majority of them agreed that the time spent and quantity of information provided during counselling was adequate, and that pharmacy waiting times were not too long. Many of them also agreed that dispensers were friendly, used language(s) or terms during counselling that they could understand and that counselling areas were comfortable and offered adequate

privacy. Other hospital-based studies that have assessed patient satisfaction with these aspects of pharmacy care have largely also reported similar findings [10, 30–34], although a few of them reported patient dissatisfaction especially with comfort [31, 33], and privacy of dispensing areas [10].

In this study, the average overall satisfaction with medication counselling score was 8.6 on a 10-point scale. This could be an actual representation of the true satisfaction levels of the respondents surveyed, since there is evidence showing that several of the items they were satisfied with e.g. waiting time, content of and time spent during medication counselling as well as pharmacist attitude are all positively linked to overall patient satisfaction [9, 32]. In addition, another study that used a 10-point scale to assess overall satisfaction in hospital outpatients also reported a similarly high, although slightly lower average rating of 7.8 [32]. However, this result should also be interpreted with caution, especially due to the marked ceiling effect observed with the responses. The ceiling effect is often observed when the instrument used is not sensitive enough to discriminate between satisfaction levels amongst respondents [35].

Several respondent characteristics including gender, educational level and income were found to be associated with overall satisfaction in this study. The effect of gender on patient satisfaction is still uncertain, because while some studies have reported that female sex is generally associated with higher satisfaction levels [18, 24, 30], others have reported higher satisfaction levels in males [9, 32, 36]. As was also seen in this study, satisfaction has been shown to decrease as educational level increases [9, 18, 24, 30, 31], so a similar effect would be expected with income and this has equally been observed in another study [24]. These findings could perhaps be explained by the fact that respondents who were wealthier/better educated likely had higher expectations for service quality, whereas those lower on the socio-economic scale may have had fewer (or lower) expectations for satisfaction. Finally, respondents in this study that were counselled through the window had significantly higher satisfaction scores than those counselled in private counselling rooms or semi-private areas. This is a particularly noteworthy finding, because it is widely believed that private counselling areas provide the best avenue for patient counselling, and there is even some evidence to support this [37]. Perhaps, this finding could be linked to the actual content of the counselling provided by dispensers in those cases, which may have been of better quality than that provided in the other settings.

With respect to medication knowledge, majority of respondents correctly knew the routes and frequency of administration of the selected medication. Similar results have also been reported from studies conducted in Ethiopia [13, 16], Saudi Arabia [38], the United States [11], Lebanon [14], Portugal [12] and Spain [39]. On the other hand, just a little under half of respondents knew the correct indication and daily timing for their medication. Some studies have reported that majority of patients know the indication(s) for their medicines [7, 11–13, 16, 39], although an Egyptian study reported that while respondents had poor knowledge about the indications of their medicines, they almost all knew the correct timing of doses [15].

Over 60% of our study's respondents did not know the duration of therapy, name or any other additional information (including side effects) about the selected medication. Many studies agree that patients do not know much additional information about their medicines especially about things like side effects and other precautions [7, 11, 12, 17, 39]. It has also been established that patients often do not know what to do if they miss a dose of their medication [13, 16, 17].

As was also the case in this study, several studies have also reported important knowledge gaps in patients with regards to relevant information about various classes of medication including NSAIDs [40], artemisinin containing antimalarials [41], cardiovascular medication [42] and antibiotics [43].

Strengths of this study include the large numbers of respondents surveyed, and the fact that respondents were sampled from major public healthcare facilities (secondary and tertiary hospitals) found within the country. However, certain limitations should also be noted. Firstly, by assessing medication knowledge against our general knowledge of pharmacology, we may have inadvertently underestimated the knowledge of patients prescribed medications for off-label indications. Furthermore, the fact that the allocated numbers of patient interviews for all of the hospitals sampled from Kaduna state could not be reached might also have affected our findings. Although this limitation may be explained by the differences in population between the two states (Kano's population of 13.4 million estimated inhabitants is almost twice as large as Kaduna's estimated population of 7.7 million). As a consequence of the non-probability sampling method used, the data obtained was also treated as a single cohort, potentially obscuring any differences that might have existed between the sampled hospitals. And although the actual personnel involved in counselling (pharmacist or technician) may have affected some of our results, we were unable to control for this possible confounder. Finally, as is often the case with questionnaire based studies, social desirability bias (a tendency for research subjects to select responses they believe are more socially acceptable) cannot be totally ruled out.

## Conclusion

Respondents' experiences during medication counselling in this study were generally positive, and their satisfaction with medication counselling was high. However, they did not know much about their medication except for route and frequency of administration. These results seem to imply that many patients within the study setting do not have adequate information on how to take their medicines correctly, which could contribute to poor treatment outcomes, increasing treatment failure rates and rising antimicrobial resistance. Thus, efforts to improve medication counselling should focus on increasing patient medication knowledge However, because medication knowledge is dependent on several other factors including medication literacy, health professionals' medication knowledge etc. Further research is recommended to better understand these findings and help identify effective interventions to improve medication knowledge.

## Acknowledgments

The authors would like to acknowledge the efforts of the data collectors involved in the study.

## Author Contributions

**Conceptualization:** Samirah N. Abdu-Aguye, Kamilu S. Labaran, Nuhu M. Danjuma, Shafiu Mohammed.

**Data curation:** Samirah N. Abdu-Aguye.

**Formal analysis:** Samirah N. Abdu-Aguye, Kamilu S. Labaran, Nuhu M. Danjuma, Shafiu Mohammed.

**Investigation:** Samirah N. Abdu-Aguye.

**Methodology:** Samirah N. Abdu-Aguye, Kamilu S. Labaran, Nuhu M. Danjuma, Shafiu Mohammed.

**Supervision:** Kamilu S. Labaran, Nuhu M. Danjuma, Shafiu Mohammed.

**Validation:** Kamilu S. Labaran.

**Visualization:** Samirah N. Abdu-Aguye.

**Writing – original draft:** Samirah N. Abdu-Aguye.

**Writing – review & editing:** Samirah N. Abdu-Aguye, Kamilu S. Labaran, Nuhu M. Danjuma, Shafiu Mohammed.

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
