## [Decision Letter · Decision Letter 0]

11 Dec 2021

PONE-D-21-18329An exploratory study of outpatient medication knowledge, experiences and satisfaction with medication counselling in North-western NigeriaPLOS ONE

Dear Dr. Abdu-Aguye,

Thank you for submitting your manuscript to PLOS ONE. After careful consideration, we feel that it has merit but does not fully meet PLOS ONE’s publication criteria as it currently stands. Therefore, we invite you to submit a revised version of the manuscript that addresses the points raised during the review process.

You will note there is only 1 peer reviewer, instead of the usual 2.  I had significant difficulty finding suitable reviewers and will have to proceed based on only a single reviewer's plus my own comments.  In addition to the comments from the reviewer, please note the following:Who actually collected the data?  What was their training to perform data collection?You treat all of the data as a single cohort.  Please comment on potential differences in results had you stratified by hospital.Counseling may be performed by a pharmacist or a technician.  You do not make this distinction which may influence results.Your use of a general score for satisfaction is problematic.  As noted, results tended to cluster around 10.  Is this due to actual satisfaction?  Or is it a function of patients not really knowing what to expect and just assigning a 10.  Without a better definition of what satisfaction consisted above, perhaps patients are rating primarily speed of service, cost of medication or some other confounder.When did the 4 pharmacists meet in conference to discuss the correctness of patients' answers?  Was this concurrent with data collection or at another time?Why were better educated/wealthier patients less likely to be satisfied?  I relate this to point 4 above - perhaps they had actual expectations for the quality of service, whereas those lower on the socio-economic scale may have had fewer (or at least different) expectations for satisfaction. 

We look forward to receiving your revised manuscript.

Kind regards,

John Rovers, PharmD, MIPH

Academic Editor

PLOS ONE

Journal Requirements:

2. Please provide additional details regarding participant consent. In the ethics statement in the Methods and online submission information, please describe 1) whether verbal consent was informed consent, and 2) how verbal consent was documented and witnessed. If your study included minors, state whether you obtained consent from parents or guardians."

3. Please include additional information regarding the survey or questionnaire used in the study and ensure that you have provided sufficient details that others could replicate the analyses. For instance, if you developed the survey or questionnaire as part of this study and it is not under a copyright more restrictive than CC-BY, please include a copy, in both the original language and English, as Supporting Information. If the questionnaire is published, please provide a citation to the (1) questionnaire and/or (2) original publication associated with the questionnaire.

Reviewers' comments:

Reviewer's Responses to Questions

**Comments to the Author**

1. Is the manuscript technically sound, and do the data support the conclusions?

Reviewer #1: Partly

2. Has the statistical analysis been performed appropriately and rigorously? 

Reviewer #1: No

3. Have the authors made all data underlying the findings in their manuscript fully available?

Reviewer #1: Yes

4. Is the manuscript presented in an intelligible fashion and written in standard English?

Reviewer #1: Yes

5. Review Comments to the Author

Reviewer #1: General comments:

The authors aim to evaluate exit-knowledge of dispensed medicines and counseling service satisfactions among outpatients served at selected hospital pharmacies in Northwestern Nigeria using a cross-sectional survey at exit from drug outlets. Clients’ knowledge of dispensed medicines and their satisfaction with pharmacy services can impact their manner of adherence to the received drugs. Understanding these aspects can help the pharmacy service providers to maintain their strong sides while also striving to improve their downsides related to medication counseling services. In these senses, this study has value, but it lacks clarity on how samples were determined and how the study participants (outpatients/caregivers) were selected from clients who contacted the main hospital pharmacy outlets considered. Other more detailed concerns and specific comments are highlighted as below:-

Title:

The title of this research didn't exactly reflect what had been measured by the authors. The authors conducted exit-interviews to assess about knowledge status of dispensed medications among outpatients/caregivers and their satisfaction with the counseling services they obtained from pharmacy professionals at eight main outpatient pharmacy units in Northwestern Nigeria. The experiences of the respondents during medication counseling service that authors assessed had helped them rate the counseling service satisfaction and it need not stand alone. Accordingly, exit-knowledge of dispensed medicines and satisfactions with medication counseling service among outpatients served at selected hospital pharmacies in Northwestern Nigeria can be specifically reflected.

Abstract:

Objective subsection of this section has an ambiguity. I think the study participants' medication experience during counseling that authors assessed was part of satisfaction with the pharmacy services that included medication counseling. Instead, they assessed medication knowledge of the participants at exit from 8 main hospital pharmacies in Northwestern Nigeria. And, this was intended to quickly test the effect of medication counseling on knowledge status of outpatients/caregivers specifically on the dispensed medicines. So, the experiences of outpatients during counseling that helped rate their satisfaction can be incorporated into the counseling service satisfaction. This needs correction across the manuscript.

Line 41, bold phrase need editorial correction.

Introduction:

Background information presented in lines 65-67 needs reference citation.

Methods:

Sample size determination and sampling technique....No probability sample was determined for this study and the authors didn't justify the reason why they dropped to do this. Again, although the authors considered non-proportional quota allocation of 100 outpatients/caregivers to each of the eight main hospital pharmacies studied in Northwestern Nigeria, they didn't explain their method of differentiation for outpatients from inpatients before enrolling them in the exit-interviews. Besides, they didn't clearly describe the method to enroll each of the outpatients (outpatients or caregivers) for the interview. Did all the outpatients/caregivers contacting the hospital pharmacies with prescriptions are interviewed about their medication knowledge, counseling status, and pharmacy service satisfaction? I wonder detail explanations on how the authors estimated 800 outpatients/caregivers and how were they selected each of the participant from among all contacts to hospital pharmacies.

Data analysis.....Authors need to explain their method on how they linked exit interviews of outpatients/caregivers with prescription reviews they did to obtain duration and frequency of use for the dispensed medicine in data collection section of the methods. Again, it is clear that non-parametric tests give crude hints of associations between exposure and outcome variables. However, it is not clear why did the authors employ non-parametric tests to assess associations between characteristics of respondents and their status of medication counseling satisfaction at exit from hospital pharmacies? I wonder how the authors could be sure on the association with this crude tests without further analytic tests.

Results:

The authors relate overall satisfaction of respondents with medication counseling service using mean score found by non-parametric tests, but it is not clear which category of the characteristics had the actual association with the satisfaction outcome. For example, the characteristics.... 'Highest educational level completed' ...has five categories and the association that the authors tested shows statistical significance, but for which category did this occur was not clear. So, I wonder why authors preferred this statistical test for associations between variables.

In table 4, authors present outpatients'/caregivers' knowledge of medicines they received from hospital pharmacies, but they didn't give summary measure of overall knowledge status for the variables considered. Better to include this in results section while also explaining ways how this summary measure was conducted in methods section of the manuscript. In the same table, responses look mismatched for name of medication categories. Please check this.

Discussion:

The first paragraph of the discussion shall indicate a brief overview of the key findings for the study that are to be interpreted one by one in next paragraphs. Instead, the authors restated their objective that needs correction. Again, ambiguity in this objective statement also needs correction to reflect what was exactly measured.

In discussion, I think the outpatients/caregivers participated in this exit-interview study rated their medication counseling satisfaction based on their experience about the service during the counseling. And, this experience of the respondents about their medication counseling need to be integrated into satisfaction with the medication counseling service. Moreover, the authors didn't explain details of why more educated respondents were less satisfied about the dispensed medication counseling service provided to them. By the same token, authors should well explain why a relatively reach respondents were less satisfied.

Conclusion:

Some aspect of this conclusion looks general whilst few other key findings were omitted from these key message statements.

6. PLOS authors have the option to publish the peer review history of their article (what does this mean?). If published, this will include your full peer review and any attached files.

Reviewer #1: No

---

## [Author Response · Author response to Decision Letter 0]

15 Jan 2022

Dear Editor and Reviewer.

Thank you so much for taking your time to review our paper and offer constructive input. Please find below our responses with reason(s) where relevant to the comments raised during the review. Thank you again.

Editors’ comment: Who actually collected the data? What was their training to perform data collection?

Action(s) taken, with reason(s) where relevant: Three young pharmacists with 1-2 years’ work experience served as the research assistants who collected the data. They were trained over a three-day period on exit interviewing by the principal investigator, who also sat through the first 20 interviews conducted by each of these assistants. This information has now been included in lines 149-152.

Editors’ comment: You treat all of the data as a single cohort. Please comment on potential differences in results had you stratified by hospital.

Action(s) taken, with reason(s) where relevant: Because of the non-probability sampling method used to allocate sample sizes, it was not possible to accurately compare findings between the hospitals sampled. This is a limitation of the study and has now been included in lines 381-383. 

Editors’ comment: Counseling may be performed by a pharmacist or a technician. You do not make this distinction which may influence results.

Action(s) taken, with reason(s) where relevant: This is also true. This has now also been included as a study limitation (Lines 383-385).

Editors’ comment: Your use of a general score for satisfaction is problematic. As noted, results tended to cluster around 10. Is this due to actual satisfaction? Or is it a function of patients not really knowing what to expect and just assigning a 10. Without a better definition of what satisfaction consisted above, perhaps patients are rating primarily speed of service, cost of medication or some other confounder

Action(s) taken, with reason(s) where relevant: This is noted and has been discussed in lines 329 - 338.

Editors’ comment: When did the 4 pharmacists meet in conference to discuss the correctness of patients' answers? Was this concurrent with data collection or at another time?

Action(s) taken, with reason(s) where relevant: These meetings were conducted after data collection had been concluded. This information has now been included in line 174

Editors’ comment: Why were better educated/wealthier patients less likely to be satisfied? I relate this to point 4 above - perhaps they had actual expectations for the quality of service, whereas those lower on the socio-economic scale may have had fewer (or at least different) expectations for satisfaction.

Action(s) taken, with reason(s) where relevant: This is noted with thanks. This information has now been included in lines 345 – 348.

Editors’ comment: Please describe 1) whether verbal consent was informed consent, and 2) how verbal consent was documented and witnessed. 

Action(s) taken, with reason(s) where relevant: Verbal consent was considered to be informed consent, and each participant was asked to provide consent before the interviews were conducted. This information has now been included in lines 196-198. 

Editors’ comment: Please include additional information regarding the survey or questionnaire used in the study and ensure that you have provided sufficient details that others could replicate the analyses. For instance, if you developed the survey or questionnaire as part of this study and it is not under a copyright more restrictive than CC-BY, please include a copy, in both the original language and English, as Supporting Information

Action(s) taken, with reason(s) where relevant: A copy of the questionnaire has now been uploaded as an appendix.

Reviewers’ comment (Title): The title of this research didn't exactly reflect what had been measured by the authors. The authors conducted exit-interviews to assess about knowledge status of dispensed medications among outpatients/caregivers and their satisfaction with the counseling services they obtained from pharmacy professionals at eight main outpatient pharmacy units in Northwestern Nigeria. The experiences of the respondents during medication counseling service that authors assessed had helped them rate the counseling service satisfaction and it need not stand alone. Accordingly, exit-knowledge of dispensed medicines and satisfactions with medication counseling service among outpatients served at selected hospital pharmacies in Northwestern Nigeria can be specifically reflected.

Action(s) taken, with reason(s) where relevant: This is noted. The papers’ title has been changed and now reads as “An Exploratory Study of Outpatient Medication Knowledge and Satisfaction with Medication Counselling at Selected Hospital Pharmacies in Northwestern Nigeria”

Reviewers’ comment (Abstract): I think the study participants' medication experience during counseling that authors assessed was part of satisfaction with the pharmacy services that included medication counseling. So, the experiences of outpatients during counseling that helped rate their satisfaction can be incorporated into the counseling service satisfaction. This needs correction across the manuscript.

Action(s) taken, with reason(s) where relevant: This is also noted and the correction has been effected across the manuscript. 

Reviewers’ comment (Introduction): Background information presented in lines 65-67 needs reference citation.

Action(s) taken, with reason(s) where relevant: This has now been included (line 89). 

Reviewers’ comment (Methods): 

i. Nonprobability sampling was determined for this study and the authors didn't justify the reason why they dropped to do this. I wonder detail explanations on how the authors estimated 800 outpatients/caregivers and how were they selected each of the participant from among all contacts to hospital pharmacies.

ii. Again, although the authors considered non-proportional quota allocation of 100 outpatients/caregivers to each of the eight main hospital pharmacies studied in Northwestern Nigeria, they didn't explain their method of differentiation for outpatients from inpatients before enrolling them in the exit-interviews. 

iii. Besides, they didn't clearly describe the method to enroll each of the outpatients (outpatients or caregivers) for the interview. Did all the outpatients/caregivers contacting the hospital pharmacies with prescriptions are interviewed about their medication knowledge, counseling status, and pharmacy service satisfaction?

Action(s) taken, with reason(s) where relevant: 

i. The reason why non-probability sampling was used has been outlined in lines 120-127.

ii. Before any patient/caregiver was interviewed for the study, the data collectors asked a question to confirm that they were outpatients. This information is included in line 153

iii. Study participants were randomly sampled for these interviews. This information is provided in lines 127 and 152.

Reviewers’ comment (Methods): 

i. Authors need to explain their method on how they linked exit interviews of outpatients/caregivers with prescription reviews they did to obtain duration and frequency of use for the dispensed medicine in data collection section of the methods.

ii. Again, it is clear that non-parametric tests give crude hints of associations between exposure and outcome variables. However, it is not clear why did the authors employ non-parametric tests to assess associations between characteristics of respondents and their status of medication counseling satisfaction at exit from hospital pharmacies? 

iii. I wonder how the authors could be sure on the association with this crude tests without further analytic tests.

Action(s) taken, with reason(s) where relevant: 

i. This information is now included in lines 159-162

ii. Nonparametric tests were used because of the nature of the data collected (i.e., the data was not normally distributed).

iii. Further tests (Dunn-Bonferroni posthoc test) were conducted to identify the actual areas where these associations were seen. This information is now included in lines 187-189

Reviewers’ comment (results): 

i. The authors relate overall satisfaction of respondents with medication counseling service using mean score found by non-parametric tests, but it is not clear which category of the characteristics had the actual association with the satisfaction outcome. For example, the characteristics.... 'Highest educational level completed' ...has five categories and the association that the authors tested shows statistical significance, but for which category did this occur was not clear. So, I wonder why authors preferred this statistical test for associations between variables.

ii. In table 4, authors present outpatients'/caregivers' knowledge of medicines they received from hospital pharmacies, but they didn't give summary measure of overall knowledge status for the variables considered. Better to include this in results section while also explaining ways how this summary measure was conducted in methods section of the manuscript. 

iii. In the same table, responses look mismatched for name of medication categories. Please check this.

Action(s) taken, with reason(s) where relevant: 

i. Information about the specific categories where the statistically significant differences were obtained has now been included in lines 239-251.

ii. We did not include a summary measure of overall medication knowledge because of the wide range of medications included in our knowledge assessment. We felt (and still feel) that it was not proper to pick out some arbitrary figure/level just to classify respondents as ‘knowledgeable’ or ‘poorly knowledgeable’. Instead, reporting the data as it is would help us highlight specific knowledge gaps that respondents had, identifying potential areas to be included in future intervention design.

iii. This has now been corrected (Table 4)

Reviewers’ comment (Discussion): 

i. The first paragraph of the discussion shall indicate a brief overview of the key findings for the study that are to be interpreted one by one in next paragraphs. Instead, the authors restated their objective that needs correction. 

ii. Again, ambiguity in this objective statement also needs correction to reflect what was exactly measured.

iii. The authors didn't explain details of why more educated respondents were less satisfied about the dispensed medication counseling service provided to them. By the same token, authors should well explain why a relatively reach respondents were less satisfied. 

Action(s) taken, with reason(s) where relevant: 

i. This has now been corrected (see lines 309-318)

ii. This objective has now been reworded as earlier recommended (lines 309-310)

iii. This explanation has now been provided in lines 345-348.

Reviewers’ comment (Conclusion): Some aspect of this conclusion looks general whilst few other key findings were omitted from these key message statements.

Action(s) taken, with reason(s) where relevant: The conclusion has now been reworded.

---

## [Decision Letter · Decision Letter 1]

14 Feb 2022

PONE-D-21-18329R1An Exploratory Study of Outpatient Medication Knowledge and Satisfaction with Medication Counselling at Selected Hospital Pharmacies in Northwestern NigeriaPLOS ONE

Dear Dr. Abdu-Aguye,

Thank you for submitting your manuscript to PLOS ONE. After careful consideration, we feel that it has merit but does not fully meet PLOS ONE’s publication criteria as it currently stands. Therefore, we invite you to submit a revised version of the manuscript that addresses the points raised during the review process.

As noted by the reviewer, one additional change to your Results Section should still be addressed

We look forward to receiving your revised manuscript.

Kind regards,

John Rovers, PharmD, MIPH

Academic Editor

PLOS ONE

Journal Requirements:

Reviewers' comments:

Reviewer's Responses to Questions

**Comments to the Author**

1. If the authors have adequately addressed your comments raised in a previous round of review and you feel that this manuscript is now acceptable for publication, you may indicate that here to bypass the “Comments to the Author” section, enter your conflict of interest statement in the “Confidential to Editor” section, and submit your "Accept" recommendation.

Reviewer #1: All comments have been addressed

2. Is the manuscript technically sound, and do the data support the conclusions?

Reviewer #1: Yes

3. Has the statistical analysis been performed appropriately and rigorously? 

Reviewer #1: I Don't Know

4. Have the authors made all data underlying the findings in their manuscript fully available?

Reviewer #1: Yes

5. Is the manuscript presented in an intelligible fashion and written in standard English?

Reviewer #1: Yes

6. Review Comments to the Author

Reviewer #1: One thing (in results), the authors didn't address a summary measure of overall medication knowledge for a reason of wide range medications they assessed. However, this is not about picking a random number/figure as they claimed. They are supposed to evaluate correctness of outpatients for each item testing knowledge of users/guardians about the medicines they received. In this regard, correct answers of the users for all questions assessing knowledge can be evaluated for a summary measure description of the knowledge. Otherwise, it is doubtful about their systematic assessment of the clients' knowledge at exit from drug outlets.

7. PLOS authors have the option to publish the peer review history of their article (what does this mean?). If published, this will include your full peer review and any attached files.

Reviewer #1: No

---

## [Author Response · Author response to Decision Letter 1]

20 Mar 2022

Dear Reviewer.

Thank you so much for taking your time to review our paper and offer constructive input. Please find below our response to the comment raised during the second review. Thank you again.

Reviewers’ comment: One thing (in results), the authors didn't address a summary measure of overall medication knowledge for a reason of wide range medications they assessed. However, this is not about picking a random number/figure as they claimed. They are supposed to evaluate correctness of outpatients for each item testing knowledge of users/guardians about the medicines they received. In this regard, correct answers of the users for all questions assessing knowledge can be evaluated for a summary measure description of the knowledge. Otherwise, it is doubtful about their systematic assessment of the clients' knowledge at exit from drug outlets.

Action(s) taken, with reason(s) where relevant: This is noted. A summary measure of overall medication knowledge has now been included in the manuscript (see lines 263 to 264). A few lines have also now been added to the methods section (lines 185-191) describing how the summary measure was calculated.

---

## [Editor Report · Decision Letter 2]

28 Mar 2022

An Exploratory Study of Outpatient Medication Knowledge and Satisfaction with Medication Counselling at Selected Hospital Pharmacies in Northwestern Nigeria

PONE-D-21-18329R2

Dear Dr. Abdu-Aguye,

We’re pleased to inform you that your manuscript has been judged scientifically suitable for publication and will be formally accepted for publication once it meets all outstanding technical requirements.

Kind regards,

John Rovers, PharmD, MIPH

Academic Editor

PLOS ONE
---

## [Editor Report · Acceptance letter]

31 Mar 2022

PONE-D-21-18329R2 

AN EXPLORATORY STUDY OF OUTPATIENT MEDICATION KNOWLEDGE AND SATISFACTION WITH MEDICATION COUNSELLING AT SELECTED HOSPITAL PHARMACIES IN NORTHWESTERN NIGERIA 

Dear Dr. Abdu-Aguye:

I'm pleased to inform you that your manuscript has been deemed suitable for publication in PLOS ONE. Congratulations! Your manuscript is now with our production department. 

Kind regards, 

on behalf of

Dr. John Rovers 

Academic Editor

PLOS ONE